# Pinpointed Stimulation of EphA2 Receptors via DNA-Templated Oligovalence

**DOI:** 10.3390/ijms19113482

**Published:** 2018-11-06

**Authors:** Christin Möser, Jessica S. Lorenz, Martin Sajfutdinow, David M. Smith

**Affiliations:** 1DNA Nanodevices Unit, Department Diagnostics, Fraunhofer Institute for Cell Therapy and Immunology IZI, 04103 Leipzig, Germany; christin.moeser@izi.fraunhofer.de (C.M.); jessica.lorenz@izi.fraunhofer.de (J.S.L.); martin.sajfutdinow@izi.fraunhofer.de (M.S.); 2Institute of Biochemistry and Biology, Faculty of Science, University of Potsdam, 14476 Potsdam, Germany; 3Peter Debye Institute for Soft Matter Physics, Faculty of Physics and Earth Sciences, University of Leipzig, 04103 Leipzig, Germany; 4Fraunhofer Project Center “Microelectronic and Optical Systems for Biomedicine” (MEOS), 99099 Erfurt, Germany

**Keywords:** DNA nanostructure, ephrin, EphA2, SWL, PC-3 cells, multivalence

## Abstract

DNA nanostructures enable the attachment of functional molecules to nearly any unique location on their underlying structure. Due to their single-base-pair structural resolution, several ligands can be spatially arranged and closely controlled according to the geometry of their desired target, resulting in optimized binding and/or signaling interactions. Here, the efficacy of SWL, an ephrin-mimicking peptide that binds specifically to EphrinA2 (EphA2) receptors, increased by presenting up to three of these peptides on small DNA nanostructures in an oligovalent manner. Ephrin signaling pathways play crucial roles in tumor development and progression. Moreover, Eph receptors are potential targets in cancer diagnosis and treatment. Here, the quantitative impact of SWL valency on binding, phosphorylation (key player for activation) and phenotype regulation in EphA2-expressing prostate cancer cells was demonstrated. EphA2 phosphorylation was significantly increased by DNA trimers carrying three SWL peptides compared to monovalent SWL. In comparison to one of EphA2’s natural ligands ephrin-A1, which is known to bind promiscuously to multiple receptors, pinpointed targeting of EphA2 by oligovalent DNA-SWL constructs showed enhanced cell retraction. Overall, we show that DNA scaffolds can increase the potency of weak signaling peptides through oligovalent presentation and serve as potential tools for examination of complex signaling pathways.

## 1. Introduction

The field of structural DNA nanotechnology is based on using DNA as construction material for building nanometer-scale objects [1,2,3,4,5,6]. Stability, specific base pairing and the biocompatibility of DNA molecules are beneficial properties that have strengthened the idea of building discrete nanostructures according to the diverse methods of DNA self-assembly and using them for a variety of applications. To equip DNA-based objects with well-defined roles, it is possible to attach functional molecules (e.g., fluorophores, peptides and aptamers) internally or on the ends of the constituent DNA strands via different chemical reactions [7]. Since the underlying nucleotide sequence, and thereby the exact molecular structure of these DNA-based objects can be determined during the design process, they can be used as scaffolds to create precise geometric arrangements of conjugated molecules with single-nanometer resolution. In the case of double-stranded DNA, a single base pair corresponds to a 0.34 nm rise along the helical axis, facilitating a more precise spatial resolution for the placement of single molecules than is practically available through other means of top-down lithography or bottom-up molecular programming. This is particularly relevant for interacting with and even controlling the behaviors of biological systems, since, by attaching several ligands to DNA objects, one can achieve optimized binding and activation of target structures such as proteins and receptors by matching their naturally defined distances between different binding or active sites. Consequently, DNA nanostructures have been studied as platforms for therapeutic agents, particularly anti-cancer compounds [8,9,10,11,12,13,14].

Erythropoietin-producing hepatocellular carcinoma (Eph) receptors are receptor tyrosine kinases (RTKs) that are activated by Eph family receptor interacting protein (ephrin) ligands. The Eph family receptors can be divided into two subclasses: EphA and EphB whereas the first is mainly bound by ephrin-A ligands and the second by ephrin-B ligands [15,16]. Since both receptor and ligand are located on the extracellular side of the cell membrane, the contact of two adjacent cells is necessary for interaction and activation [17]. A special feature of Eph-ephrin signaling is bidirectional signal transduction in both Eph- (“forward” signaling) and ephrin-presenting cells (“reserve” signaling). Ephrin pathways are important for angiogenesis, adult tissue homeostasis, embryogenesis, and other developmental processes. However, they are additionally key players in many pathological conditions; for example, the EphA2 receptor is widely upregulated in many cancer types (breast [18], prostate [19], ovarian [20], lung [21]). EphA2 overexpression is linked to poor clinical outcome and malignant progression, but those properties are most likely independent of the ligand binding to the receptor, and instead result from crosstalk between EphA2 and Akt [22,23]. Interestingly, Eph receptors can also act as tumor suppressors when activated by ephrin or ephrin-like ligands. Downstream signaling correlates with inhibition of cell proliferation, migration, invasiveness and adhesion, which are regulated via RAS-ERK [24], Akt-mTORC1 [25] and several integrin-dependent pathways [26]. As a result, Eph receptors and ephrins are increasingly studied as potential therapeutic targets [27,28]. In addition, these receptors are internalized upon ligand binding and their expression in normal tissues is low compared to cancerous tissue, making them suitable to specifically deliver anti-cancer agents [29,30] or serve as specific tissue markers for imaging agents [31,32].

The agonistic ephrin-mimicking peptide SWL was previously discovered by phage display and was shown to specifically bind EphA2 receptors on their ligand-binding domain, thereby activating downstream signaling pathways [33,34]. By presenting two covalently conjugated SWL peptides as dimer, the binding to EphA2 receptors could be enhanced more than 10-fold [35]. However, this was accompanied with the disadvantage of a decreased half-life compared to the SWL monomer, hindering thorough assessment of cooperative binding in cellular systems. In contrast to the natural ephrin ligands, which are considered to be promiscuous due to their propensity to bind and stimulate multiple Eph receptors [36], SWL’s complete specificity to EphA2 eliminates the possibility of downstream interference reported to occur as a result of simultaneously triggering competing pathways [37,38].

Here, we adopt a DNA-templated approach, where we bind up to three SWL peptide monomers to stable DNA nanostructures and examine the influence of this oligovalent presentation on PC-3 prostate cancer cells that overexpress the EphA2 receptor. The principle of oligovalent presentation and binding is ubiquitously found in nature, being, among others, the basis of specific cell-cell interactions and pathogen-host recognition [39]. It is based on several simultaneously occurring interactions that collectively are stronger than one single interaction would be. Upon binding of the dimeric ephrin ligand to a pair of Eph receptors, their dimerization is followed by autophosphorylation of tyrosine residues in the cytosolic domain of the receptors and subsequent activation of signaling pathways. In contrast to most RTKs, downstream Eph-ephrin signaling requires the formation of higher-order clusters [40]. Since the association of as few as three ligand-receptor complexes has been shown to cause oligomerization [41,42] a three-valent DNA template called the “DNA trimer” was chosen for this study (Figure 1). It consists of three partially complementary strands, has a molecular mass of approximately 28 kDa when unmodified, and of approximately 34 kDa when modified with three monomeric SWL peptides. A related DNA-based approach to activate EphA2 signaling pathways by controlling cluster proximity was introduced by Shaw et al. in 2014 [8]. They engineered a large DNA origami structure (approximately 5 MDa in size) that positions pairs of dimeric ephrin-A5 protein ligands at two different distances from each other, 43 and 100 nm. Here, we chose the comparatively small SWL peptide (1.7 kDa) instead of natural ephrin protein ligands (e.g., ephrin-A1 with molecular weight (MW) approx. 21 kDa) since these mimic the binding and stimulation activity of the full ligand, albeit with specificity to EphA2, while still allowing the exploitation of sub-nanometer spatial resolution that DNA can provide [43]. When conjugated at the ends of the arms on the DNA trimer, SWL peptides are approximately 9 nm apart when the structure is in outstretched conformation. Although the structure’s arms are rigid on these length scales (DNA persistence length is 50 nm), the junction in the middle supplies the DNA trimers with needed flexibility for adjusting to the exact conformation of the EphA2 ligand-binding domain. Furthermore, nanostructures built from DNA or other modified nucleotide variants are very stable molecules [44] and therefore can be more suitable to survive in a biological environment than peptides alone [35] (Figure A4). To place three SWL peptides onto the DNA trimer, *N*-hydroxysuccinimide (NHS) ester reactions as well as copper-free click chemistry reactions [45,46,47,48,49] were performed (Figure 1b) to attain nearly quantitative yields.

The specific binding of SWL-coupled DNA trimers to PC-3 prostate cancer cells known to overexpress EphA2 was evaluated via ligand titration in in flow cytometry experiments. Subsequently, their ability to activate these receptors via phosphorylation was evaluated for relative efficacy and potency. As a result of activation, cells undergo characteristic morphological changes, where they “round up” and retract their periphery. We further characterized the assembly and purity of SWL-DNA trimers, investigated their stability in serum-containing medium and when incubated together with cells.

## 2. Results

### 2.1. Binding of SWL-DNA Trimers to PC-3 Cells

PC-3 cells are known to overexpress EphA2 receptors [19] and to confirm this, we initially probed its expression using a phycoerythrin (PE) labeled anti-human EphA2 antibody in flow cytometry experiments. As a negative control HL-60 cells, which should only express these receptors after maturation [51], were tested and the expected lack of expression was confirmed compared to PC-3 cells (Figure A5). EphA2 expressing PC-3 cells were then treated with different concentrations of DNA trimers carrying both a Cy3 dye and between 0–3 SWL peptides and were subsequently analyzed via flow cytometry (Figure 2). As expected, binding curves revealed that the construct with three SWL peptides (3xSWL-DNA) displays highest binding activity. In comparison, the binding activity of structures carrying one SWL peptide is nearly indistinguishable from nonspecific interactions occurring between the bare DNA trimer and cells. Structures carrying two peptides do display moderate binding, although at approximately one-quarter of the levels of the three-peptide structures at the highest concentration.

For both the two- and three-peptide variants, the highest concentration of 25 µM was not sufficient to reach saturation of binding, as was particularly evident in the latter case. This lack of saturation likely results from the active internalization via endocytosis following activation of EphA2 pathways, leading to an accumulation of fluorescent molecules within the cell interior. Nevertheless, the onset of binding above 1.5 µM for the 3xSWL-DNA trimer is clear, compared to the 1xSWL-DNA and 2xSWL-DNA trimers, both of which do not show any comparable amount of binding below the maximum concentration. Due to the functional connection between receptor binding and internalization, biochemical signaling events and phenotype changes resulting from treatment with the DNA-peptide constructs were also examined.

### 2.2. Confirmation of EphA2 Pathway Activation via Receptor Phophorlyation

The efficacy and potency (EC_50_) of EphA2 receptor phosphorylation resulting from binding of the various constructs (monomeric peptide, mono-, bi- and trivalent presentation on the DNA trimer and positive/negative controls) was quantified by sandwich enzyme-linked immunosorbent assay (ELISA) detecting first EphA2 receptors and second phospho-tyrosines. To qualitatively compare the different ligands examined, a single concentration of the different ligands was analyzed first (Figure 3). These were chosen to be either near the onset of binding for the 3xSWL-DNA trimer for the DNA structures, or according to published protocols for the natural ligand and SWL. As a result of treatment, high phosphorylation signals were observed for the natural ligand ephrin-A1 (1.5 µg/mL) and the 3xSWL-DNA trimer (9 µM of the trivalent construct). Monomeric SWL peptide (150 µM) and 2xSWL-DNA trimer (9 µM of the bivalent construct) were of a comparable signal level to each other, clearly lower than the trivalent construct. However, it should be noted that the monovalent SWL peptide was applied at approximately a 16-fold higher concentration than the bivalent 2xSWL-DNA construct, clearly indicating a synergistic oligovalent effect rather than simply an additive effect due to doubling the total amount of SWL peptides in solution. As expected, negative controls of phosphate-buffered saline (PBS) or DNA trimers without any SWL peptides did not lead to any clearly enhanced phosphorylation of tyrosine residues.

Furthermore, the phosphorylation of EphA2 receptors resulting from the application of different concentrations of the ligands enabled the calculation of approximate EC_50_ values for the DNA-peptide constructs by least-square analysis (Figure 4, Equation (1)) [52]. As expected, 3xSWL-DNA trimers led to the highest phosphorylation signal (EC_50_ = 0.0190 ± 0.0046 µM), i.e., overall efficacy, followed by 2xSWL-DNA trimers (EC_50_ = 0.0572 ± 0.0113 µM) and 1xSWL-DNA trimers (EC_50_ = 2.7427 ± 1.9837 µM) for concentrations up to 30 µM. For SWL, concentrations up to 500 µM were tested and fitting revealed an EC_50_ value of 153.8287 ± 115.3869 µM. DNA trimers without any SWL (0xSWL-DNA trimer) did not evoke phosphorylation of EphA2 receptors, and accordingly the absorbance signal is within the background (approximately 0 in Figure 4a). Figure A7 presents the data obtained for the natural ligand ephrin-A1 (EC_50_ = 0.0027 ± 0.0005 µM).

### 2.3. PC-3 Cell Rounding Caused by EphA2 Activation

Phosphorylation of EphA2 receptors is required for subsequent signaling pathways in EphA2 receptor presenting cells, therefore it is expected that multivalent presentation of the peptide on the trimeric DNA construct will lead to the most significant cell phenotype changes. As a result of downstream processes resulting from EphA2 activation, PC-3 cells retract their membrane protrusions and “round up,” adopting a compact morphology [26]. The overall impact of monomeric SWL peptide or the SWL-DNA trimer displaying different numbers of peptides on cell morphology was qualitatively assessed via microscopic imaging, as seen in Figure 5. For 3xSWL-DNA trimers, a concentration of 20 µM, well above the measured EC_50_ values, was used while concentrations of the natural ligand and monovalent SWL were chosen according to previous reports.

Slight rounding of cells following starvation was a baseline effect for all samples, likely due to the lack of nutrients and medium. As expected, the most prominent differences were seen for 3xSWL-DNA trimers. At a concentration of 20 µM, rounding of cells was already observed after 20 min, and was ubiquitous in the sample after 60 min of treatment. By contrast, after 20 min a significant amount of rounding beyond the likely starvation-induced effects was not significantly clear for the 1xSWL-DNA or 2xSWL-DNA structures (Figure A8). A 150 µM concentration of the monomeric peptide showed no noticeable change in morphology after 20 min when compared to the negative controls of PBS and the DNA structure without any peptides.

In comparison to the 3xSWL-DNA structure, only moderate changes in morphology were observed to arise from the application of 1.5 µg/mL of the natural ephrin-A1 ligand after 20 and 60 min. This concentration is approximately 27 times the EC_50_ value of 2.654 nM as shown in Figure A7. Downstream impact on morphological phenotype is clearly suppressed even though the natural ligand did still lead to high levels of EphA2 phosphorylation, as seen in Figure 3 and Figure A7. While seemingly contradictory, this is consistent with some previous reports where additional stimulation of the Rho-ROCK1 signaling pathway by external serum factors is necessary to cause significant levels of cell rounding following serum starvation [53].

## 3. Discussion

Here, it was shown that even simple DNA nanostructures consisting of a few strands are not only limited to serving as functional carriers for bioactive peptides such as SWL but can also enhance their activity and trigger specific downstream signaling pathways in a pinpointed manner. These rationally designed, structural scaffolds not only hold the peptides in a controlled spatial average distribution while still maintaining flexibility at the central joint to finely adjust to the arrangement of binding sites; they also preserve their functionality and provide the basis for oligovalent binding to target structures, and thus an enhanced efficacy when compared to the monovalent peptides.

The binding of Cy3-labeled SWL-DNA trimers to PC-3 cells via EphA2 receptors was analyzed using flow cytometry to determine binding curves (Figure 2). Preparation limitations impeded the production and testing of higher concentrations of the constructs. Therefore, saturation could not be achieved, likely due to the fact that EphA2 receptors are internalized upon activation [54], leading to an accumulation of fluorescence within the cells. In this case, saturation, if reached, would likely correspond to the actively driven depletion of expressed EphA2 receptors from the surface of the cells, rather than equilibrium kinetics. Thus, half-maximal effective concentration (EC_50_) values or other binding constants could not be quantitatively determined for flow cytometry data. Nevertheless, enhanced concentration-dependent fluorescence signals for cells treated with the 3xSWL-DNA trimer compared to the 0–2 peptide variants clearly show an increased affinity due to higher-order presentation of SWL.

In a sandwich ELISA to quantify phosphorylation of tyrosine residues (Figure 3 and Figure 4), our results underline the presence of an oligovalent effect, as 3xSWL-DNA trimers are both more potent and more efficacious than monomeric SWL peptides. Interestingly, all SWL-conjugated DNA trimers are more potent than the monovalent, unconjugated SWL peptide. SWL and 1xSWL-DNA seem to have a similar efficacy although it should be noted that the maximum effect could not be reached due to practical limitations, implying that the calculated EC_50_ value could be higher. However, it is clear that 1xSWL-DNA trimers can reach similar phosphorylation signals with a much lower concentration, indicating that the presence of the DNA structure improves potency. An explanation could be that the attachment of SWL to DNA already improves its stability and/or binding properties. More specifically, the larger DNA construct could shield the attached peptide from proteases, which was indeed previously implicated as a limitation for the covalently dimerized SWL variant [35]. Furthermore, DNA is known to nonspecifically stick to the outer plasma membrane of cells, which would generally increase the chances for the DNA-peptide conjugate structures to interact with any surface receptors such as EphA2 as compared to the peptide by itself.

The 2xSWL-DNA and 3xSWL-DNA constructs have a noticeably elevated performance compared to 1xSWL-DNA and SWL, as shown in Figure 4a. The attachment of two peptides on the DNA structure drastically decreases the EC_50_, by nearly a factor of 50 compared to 1xSWL-DNA, which portends the influence of an oligovalent effect. This is further enhanced when 3 peptides, the suggested minimum for inducing Eph receptor clustering [42], are attached to the DNA structure. While the apparent EC_50_ value is only further decreased by a factor of three, the overall phosphorylation efficacy more than doubles, as compared to the monomeric peptide as well as both the mono- and bivalent DNA-peptide conjugates. The natural ligand ephrin-A1 remained most potent and efficacious when tested for EphA2 receptor phosphorylation in different concentrations (Figure A7 and Figure 4b). As expected, 0xSWL-DNA trimer did not lead to phosphorylation at any concentration, and therefore an EC_50_ value was not calculated. Moreover, comparatively low signals seen for PC-3 cells treated only with PBS confirmed that even though EphA2 receptors are indeed highly expressed (indicated by the PBS control in Figure A6) they are only slightly activated.

Even though the degree of activation from even the trivalent DNA construct is greatly surpassed by that achieved by the natural ephrin-A1 ligand (50% higher phosphorylation levels were achieved by 250× less of the natural ligand, see Figure 3), we do point out two factors which mitigate a direct comparison. First, it is expected that a short, linear, likely flexible peptide fragment will be entropically disfavored from specifically binding to and activating its target when compared to an active polypeptide region held in a rigidly defined conformation within a folded protein. Second, as noted before, ephrin-A1 is known to bind and activate most, if not all A-type Eph receptors, several of which are expressed in PC-3 cells [55]. While the natural ligand is clearly more efficient at stimulating phosphorylation of EphA2 dimers, this does not capture the extent to which cross-stimulation of other EphA receptors interferes with downstream pathways [37,38]. Indeed, recently reported findings from Singh et al., point out that different types of ephrin ligands, including the SWL peptide used here, give rise to diverse activities of EphA2 receptor signaling [56].

This second point is of particular importance when observing the effects on cell phenotype; namely the “rounding up” of cells following treatment. While the application of the full ligand in excess of its measured EC_50_ value for tyrosine phosphorylation did show some moderate signs of morphological changes after 20 min (Figure 5) consistent with previous observations on serum-starved PC-3 cells [53], this was clearly surpassed by effects resulting from application of the 3xSWL-DNA constructs. Even though visual, microscopic interpretation of cell morphology is to some extent a qualitative art, this nonetheless suggests that pinpointed stimulation with the EphA2-specific peptide along with the promotion of receptor clustering is a potent trigger of this particular phenotype change. This discrepancy with the significantly more efficient tyrosine phosphorylation by the natural ligand does support the possibility of negative feedback resulting from the cross-stimulation of different EphA receptors.

More generally, the presentation of highly specific peptide fragments on simple DNA scaffolds in the way shown here potentially provides a two-pronged tool for helping to unravel the complexity of signaling pathways involving promiscuous binding between a ligand and multiple receptors [57]. Beyond the A- and B-classes of Eph receptors, the binding of a single ligand to a set of multiple receptors (or vice versa) is a common occurrence in diverse processes such as immune recognition [58], tissue and organ development [59], programmed cell death aka apoptosis [60,61] and many others. The last example of cell apoptosis via activation by the TNF-related apoptosis-inducing ligand (TRAIL) ligand is a particularly compelling case for DNA-templated oligovalence; TRAIL binding to either the death (DR4, DR5) or decoy (DcR1, DcR2, osteoprotegerin) receptors can trigger or block apoptosis, respectively, with a homo-trimerization of the death receptors is a prerequisite for apoptosis stimulation. DNA-scaffolded presentation of short peptide epitopes known to be specific to a particular receptor [62,63] would enable a systematic examination of these pathways with a pinpointed resolution in terms of biochemical pathways and degree of multimerization in signaling events.

## 4. Materials and Methods

### 4.1. Synthesis of SWL-Coupled DNA Trimers

Three partially complementary strands (Table 1) were purchased with 5′ end amino-group from Biomers.net (Ulm, Germany) and delivered in dry and HPLC-purified form.

The strands were resuspended in water, the concentration was determined via analysis of ultraviolet-visible spectroscopy (NanoDrop ND-1000 UV/Vis spectrophotometer from PEQLAB Biotechnologie, Erlangen, Germany) by light absorption at 260 nm and the DNA strands were mixed in equimolar amounts to guarantee for optimal stoichiometry and thus high yields. DNA structures were assembled at 25 µM each strand in 1× PBS by heating the mixture to 95 °C for 2 min, hybridizing at 48 °C for 15 min and cooling down to 4 °C. For trimers carrying no, one or two peptides, amine-modified DNA strands were replaced by three, two and one unmodified strand, respectively. Unmodified DNA trimers were imaged with atomic force microscopy (AFM) (Figure A2); however, due to their small size, exact structural details were not discernible.

Folded DNA trimers were functionalized with peptides as previously described [49]. Briefly, DNA trimers were incubated with a 100-fold molar excess of DBCO-NHS esters (Jena Bioscience, Jena, Germany) in 1× PBS pH 7.4 at room temperature overnight. The next day, DBCO-coupled DNA trimers were purified from unconjugated DBCO-NHS esters by ethanol precipitation and subsequently incubated with a 20-fold molar excess of azide-containing peptide SWL (SWLAYPGAVSYRGG-Azidolysin; purchased from Peptide Specialty Laboratories, Heidelberg, Germany) overnight. Excess of SWL was removed via spin filtration using 1× PBS/10 mM MgCl_2_ and Amicon^®^-Ultra-0.5 Centrifugal Filter Devices with 10 K MWCO (Merck Chemicals, Darmstadt, Germany) which were used according to manufacturer’s instructions. Native polyacrylamide gel electrophoresis (PAGE, Figure A1) was used to analyze folding and functionalization. Matrix-assisted laser desorption/ionization—time-of-flight (MALDI-TOF) measurements were conducted to check for remains of uncoupled SWL after Amicon^®^ purification (Figure A3).

The distance between the peptides was calculated according to well-known structural properties of double-stranded DNA. It is known that one arm consists of 15 base pairs which equals 5.1 nm (0.34 nm per base). By applying cosine and assuming the angle on the junction to be 120°, we could calculate a distance of approx. 8.8 nm which can vary according to buffer and salt conditions.

### 4.2. Binding Assay

Binding of SWL-DNA trimers to PC-3 cells (purchased from ATCC (ATCC^®^ CRL-1435™; Manassas, VA, USA) was assessed in duplicates on an Intellicyt high-throughput flow cytometer. The day before assaying, 1 × 10^4^ PC-3 cells were seeded into a 96-well U-bottom plate. After serum starvation for 4 h, cells were washed with 1× PBS, treated with serial dilutions of Cy3 labeled DNA trimers conjugated to three, two, one or no peptide SWL (3xSWL-DNA trimer, 2xSWL-DNA trimer, 1xSWL-DNA trimer, 0xSWL-DNA trimer, respectively) for 30 min at room temperature. Subsequently cells were rinsed with 1× PBS, detached using 0.05% trypsin-EDTA and centrifuged at 1150× *g* for 5 min at room temperature. Cells were resuspended in 1× PBS/1% (*w*/*v*) bovine serum albumin (BSA) and analyzed by an iQue screener (Intellicyt, Albuquerque, NM, USA).

### 4.3. EphA2 Phosphorylation Assay

Levels of EphA2 phosphorylation were determined using the DuoSet IC Human Phospho-EphA2 kit (R&D systems, Minneapolis, MN, USA) which is based on a sandwich ELISA. Briefly, 2 × 10^5^ PC-3 cells were seeded three days before the assay for the comparison of ligands. For the concentration series used to determine EC_50_ values, 1 × 10^6^ PC-3 cells were seeded into 12-well culture plates one day before the assay. On the day of the experiment, cells were serum-starved for 4 h and treated with either 1xSWL-DNA trimer, 2xSWL-DNA trimer, 3xSWL-DNA trimer, natural ligand ephrin-A1, SWL peptide, DNA trimer only or 1× PBS/10 mM MgCl_2_ as negative control for 30 min at 37 °C. Cells were lysed using 250 µL/well lysis buffer (according to kit) and assayed as indicated by manufacturer’s instructions. Manifold washing was conducted after every step and the whole procedure was executed at room temperature. In brief, 96-well ELISA plates carrying specific human EphA2 capture antibodies were incubated with cell lysates of afore treated PC-3 cells. This was followed by the addition of anti-phospho-tyrosine-HRP antibodies which ultimately fulfil a colorimetric reaction by conversion of 3,3′,5,5′-Tetramethylbenzidine substrate solution for 25 min (ligands in Figure 4a) or 8 min (ephrin-A1 in Figure A7). This reaction was stopped by adding 2 N H_2_SO_4_ and absorbance was measured at 450 nm and at 520 nm (infinite M1000, Tecan, Switzerland); the latter was subtracted as reference value. To normalize values from concentration series, controls (only 1× PBS/10 mM MgCl_2_) were subtracted, too. Data was analyzed by fitting to the equation:
(1)$$A=A_{min}+\frac{\left(A_{max}-A_{min}\right)}{1+{\left(\frac{x}{EC_{50}}\right)}^{-b}}$$
where *A* is the observed absorbance, *A_min_* and *A_max_* are minimal and maximal absorbance respectively, and *b* is the Hill coefficient.

### 4.4. Cell Rounding Assays

PC-3 cells were checked for changes in cell morphology upon activation of EphA2 receptors and following signaling pathways. Briefly, sub-confluent PC-3 cells in 96-well plates were serum-starved for 4 h and then incubated for up to 1 h with either 3xSWL-DNA trimer (20 µM), natural ligand ephrin-A1 (1.5 µg/mL), SWL peptide (150 µM), DNA trimer only (20 µM) or 1× PBS/10 mM MgCl_2_ as control. Microscopic images of PC-3 cells were taken using a Leica DM IL microscope with 10× objective to assess cell contraction and rounding.

## Figures and Tables

**Figure 1 ijms-19-03482-f001:**
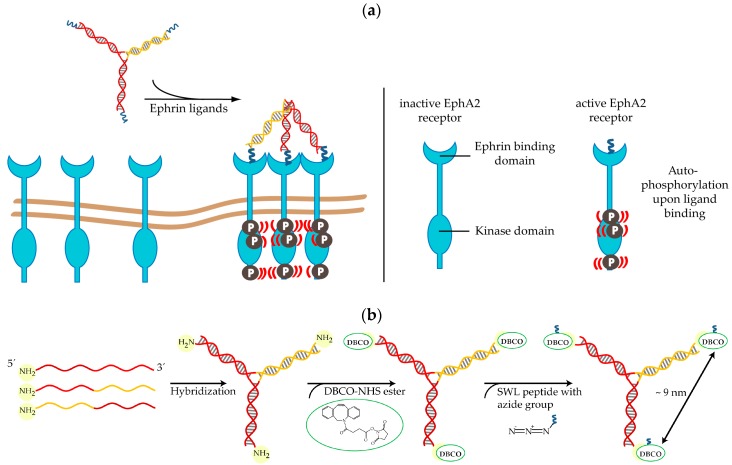
SWL-coupled DNA trimer and its action on EphA2 receptors: (**a**) EphA2 (light blue) cluster formation and subsequent autophosphorylation due to the presence of three peptides SWL attached to DNA trimers. Inactive Eph receptors are loosely distributed on cell membranes and become ordered when activated [50]. (**b**) Synthesis of SWL-coupled DNA trimer. Three partially complementary strands (complementary parts indicated by same color) are hybridized to form the DNA structure. Primary amine groups on 5′ ends react with dibenzylcyclooctyne (DBCO)-NHS esters (green) and form amid bonds. Peptides SWL with azide group on the C-terminus (dark blue) react with DBCO to form stable triazoles.

**Figure 2 ijms-19-03482-f002:**
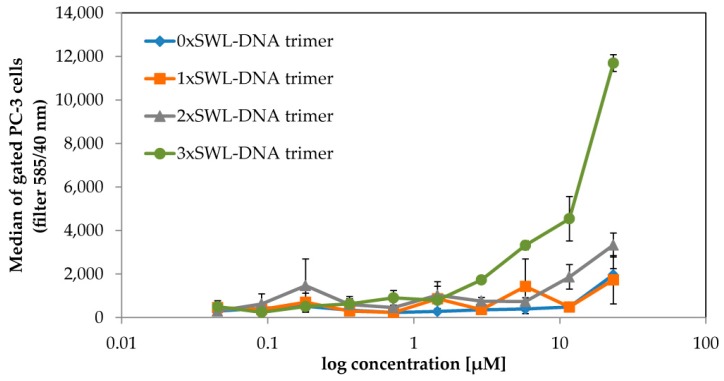
Binding of SWL-coupled DNA trimers to EphA2-expressing PC-3 cells. Serum-starved PC-3 cells were treated with SWL-DNA trimer constructs (labeled with Cy3, carrying 0-3 peptides SWL) in duplicates for 30 min at room temperature. Cells were gated and normalized by cell number. Displayed on the y-axis are average values of emission which were observed using a 585/40 nm filter (FL2-A) with standard deviation presented as error bars. The x-axis indicates concentrations of the constructs in terms of DNA. Scatter plots of raw data can be found in Figure A6.

**Figure 3 ijms-19-03482-f003:**
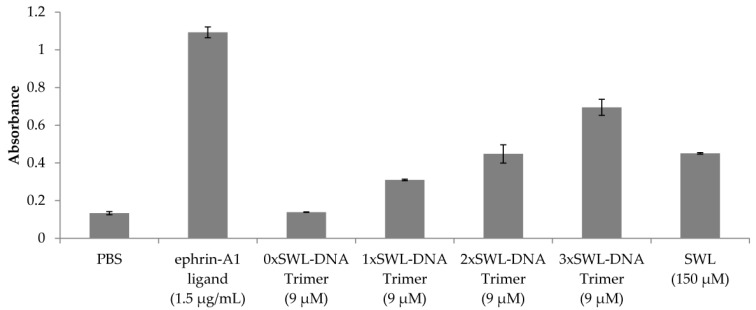
Qualitative analysis of EphA2 receptor phosphorylation. Serum-starved PC-3 cells were treated with different SWL-DNA trimer constructs (9 µM), monomeric SWL peptide (150 µM) and the natural ligand ephrin-A1 (1.5 µg/mL) in duplicates for 30 min at 37 °C. Cells were washed with 1× PBS twice and further treated according to the manufacturer’s instructions. Displayed values are averages of net absorbance (difference between absorbance at 520 nm versus 450 nm) with standard deviation presented as error bars.

**Figure 4 ijms-19-03482-f004:**
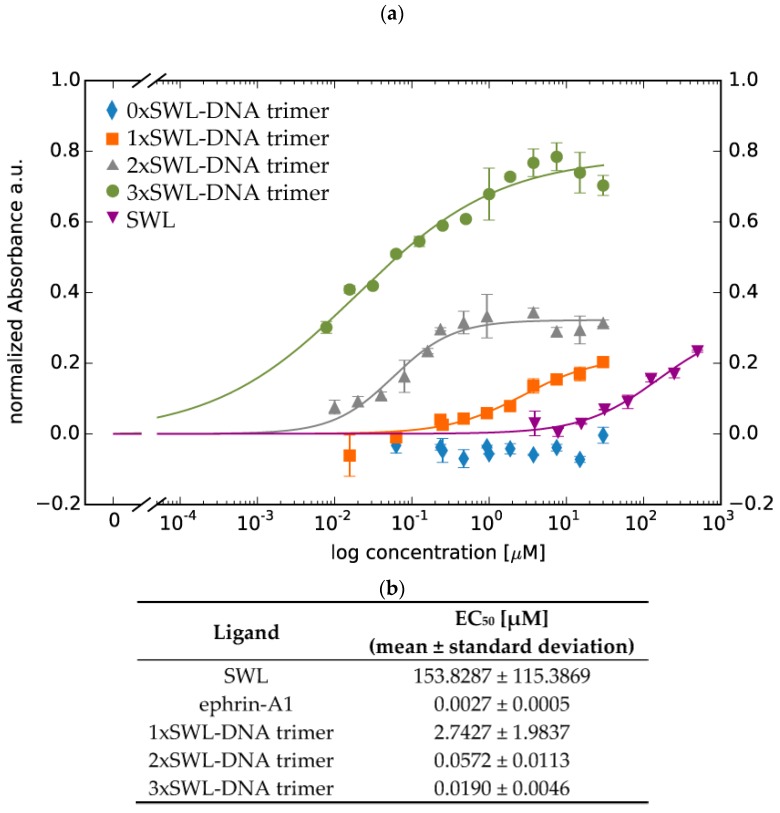
Quantitative analysis of phosphorylation of EphA2 receptors. Serum-starved PC-3 cells were treated with different concentrations of constructs in duplicates for 30 min at 37 °C. Cells were washed with 1× PBS twice and further treated according to the manufacturer’s instructions. (**a**) Displayed data points are averages of normalized net absorbance (difference between absorbance at 520 nm versus 450 nm); the net absorbance signal in the presence of different entity concentrations was normalized to the signal from treatment with 1× PBS/10 mM MgCl_2_ only (without any peptide or DNA constructs) resulting in normalized absorbance. Standard deviation is presented as error bars. The x-axis indicates concentrations of the constructs in terms of DNA for 0xSWL-DNA trimer—3xSWL-DNA trimer and the concentration of the peptide for SWL. It should be understood that the x-axis displays concentrations for whole entities not for binding entities (in this case SWL molecules). (**b**) EC_50_ values of different ligands for EphA2 receptor phosphorylation resulting from fitting (solid lines in Figure 4a, Equation (1)).

**Figure 5 ijms-19-03482-f005:**
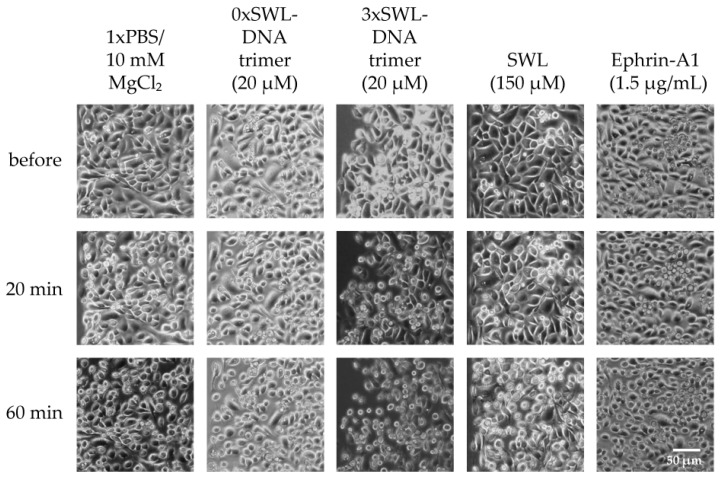
PC-3 cell rounding. Cells were seeded the day before and serum-starved for 4 h prior treatment. Samples were diluted in PBS and incubated on cells for indicated times. Cell morphology was observed and 20 and 60 min to account for inherent rounding effects from serum starvation.

**Table 1 ijms-19-03482-t001:** Oligonucleotide sequences (30 nucleotides) to form DNA trimers. Part a is complementary to a *; b to b * and c to c *. Different variations of DNA sequences were used (see footers).

Name	Sequence 5′→3′
ab	ACTATCTTTGGTCTATTATCTTGAGTCATC ^1,2,3,4^
b *c	GATGACTCAAGATAAACACACACACAACTA ^1,2^
c *a *	TAGTTGTGTGTGTGTTAGACCAAAGATAGT ^1,2^

^1^ no modification, ^2^ 5′ Aminolink C6, ^3^ 3′Cy3, ^4^ both 5**′** Aminolink C6 and 3′Cy3.

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
