# Peer review of "Pinpointed Stimulation of EphA2 Receptors via DNA-Templated Oligovalence"

_ijms, 2018, doi:10.3390/ijms19113482_

Round 1
Reviewer 1 Report
The manuscript titled “Pinpointed stimulation of EphA2 receptors via DNA-templated oligovalence” by Smith et al. describes improving the efficacy of binding of SWL to EphrinA2 by multimerizing it on a DNA nanostructure. I have following comments on the manuscript:
1. In the abstract (page 1, line 17), authors write that the efficacy of SWL was “increased by presenting numerous of these peptides on small DNA nanostructures in an oligovalent manner”. Use of the term “numerous” is misleading here since authors used a maximum of three copies of SWL.
2. On page 2, line 94, the authors mention “sub-nanometer spatial resolution that DNA can provide”. Although there have been numerous studies which have exploited the DNA structure to achieve nanometer-scale resolution in the binding of ligands, I am not aware of any studies which have been able to reach sub-nanometer level. If there has been such a study, the authors must cite that here. If not, then perhaps change language slightly to indicate what has been achieved and what is theoretically possible.
3. How did the authors arrive at the length of the arms of the DNA three-way junction? Was there any theoretical study or structural analysis behind that?
4. The authors must specify in Figure 2, the concentration values on x-axis represent the concentration of which entity.
5. On page 6, line 191, the authors mention that the DNA structure holds the peptides in a “controlled spatial distribution”. How can the DNA structure do that when it is supposed to be flexible enough to be able to reorient peptides to bind to the target?
6. The authors state that the saturation could not be achieved in Figure 2. The authors did not go higher than the concentration value which they have showed. Is it because of the cost factor or any other factor? Because, theoretically speaking, saturation could be achieved.
The authors do state that the degree of activation from the trivalent construct is greatly surpassed by that achieved by the natural ephrin-A1 ligand. However, this study is a step towards achieving the long-term goal of systematic examination and manipulation of a biochemical pathway using spatially and functionally controlled ligands. The manuscript is well written and I recommend it for publication with the aforementioned comments suitably addressed.
Author Response
Dear Reviewer,
Thank you for reading and evaluating the submitted paper.
1. In the abstract (page 1, line 17), authors write that the efficacy of SWL was “increased by presenting numerous of these peptides on small DNA nanostructures in an oligovalent manner”. Use of the term “numerous” is misleading here since authors used a maximum of three copies of SWL.
Response: The referee makes a good point, since the word "numerous" could be vague and/or misleading. We have replaced it with the more concrete term “up to three”.
2. On page 2, line 94, the authors mention “sub-nanometer spatial resolution that DNA can provide”. Although there have been numerous studies which have exploited the DNA structure to achieve nanometer-scale resolution in the binding of ligands, I am not aware of any studies which have been able to reach sub-nanometer level. If there has been such a study, the authors must cite that here. If not, then perhaps change language slightly to indicate what has been achieved and what is theoretically possible.
Response: The referee raises a valid point here since we did not provide any citation or other justification for our statement. Practically, the argument could be made that a single base-pair in dsDNA corresponds to a length of 3.4Å. This is of course difficult to experimentally prove, and this sort of precision would be subject to thermal fluctuations. However, a recent study did demonstrate that the average position of molecules could be controlled by DNA structures down to a fraction of a nanometer. Specifically, the following study was added: Funke, J. J.; Dietz, H. Placing molecules with Bohr radius resolution using DNA origami. Nat. Nanotechnol. 2016, 11, 47–52, doi:10.1038/nnano.2015.240. The authors report stepwise adjustments of distances between molecules from 1.5 to 9 nm, whereas the smallest step was 0.04 nm.
3. How did the authors arrive at the length of the arms of the DNA three-way junction? Was there any theoretical study or structural analysis behind that?
Response: The lengths of the arms were calculated according to known structural constants for dsDNA. It is known from our design selection that one arm consists of 15 base pairs which, equals 5.1 nm (0.34 nm per base). This explanation as well as the calculation for the distances between the peptides was added to 4.1.
4. The authors must specify in Figure 2, the concentration values on x-axis represent the concentration of which entity.
Response: We fully agree with the referee, and added a sentence in the caption that the x-axis represents DNA concentrations.
5. On page 6, line 191, the authors mention that the DNA structure holds the peptides in a “controlled spatial distribution”. How can the DNA structure do that when it is supposed to be flexible enough to be able to reorient peptides to bind to the target?
Response: We appreciate the tip for clarifying our point. That statement was changed to “controlled spatial average distribution with flexibility to adjust” to explain it more precisely.
6. The authors state that the saturation could not be achieved in Figure 2. The authors did not go higher than the concentration value which they have showed. Is it because of the cost factor or any other factor? Because, theoretically speaking, saturation could be achieved.
Response: We did not increase concentrations largely due to preparation and cost limitations. Additionally, as we point out in the text, saturation as defined by binding kinetics is conflated in this case with the tendency for activated receptors to become internalized. This would lead to an overall increase in the apparent saturation more related to active depletion of internalized EphA2 receptors from the outer cell surface. Since the primary aim of this experiment was to compare the binding and subsequent activation of EphA2 receptors due to DNA-peptide constructs with different numbers of peptides, we choose instead to emphasize the newly added EC50 values for EphA2 tyrosine phosphorylation as the primary quantitative measure.
Reviewer 2 Report
In this manuscript, Smith and coworkers demonstrated a multivalent DNA nanoparticle for the delivery of EphA2 ligand. The trivalent nanoparticle could deliver multiple ligands and induce higher anticancer efficacy. This is an interesting work but it needs further improvement before being accepted for publication.
1. There are not so much data in the main text, it is unnecessary to create a supplementary file for the rest of the data.
2. Natural ephrin ligands are promiscuous, then why trivalent DNA nanostructure was chosen for this study?
3. It seems discussion for the preparation of the DNA nanostructure was ignored in the manuscript, which is odd since this paper is about this DNA nanostructure. Furthermore, it is better to do an AFM characterization of the DNA nanostructure.
4. Figure 3 showed only qualitative data of anticancer efficacy, it is better to do more concentration gradients and calculate the IC50 of each formulation.
Author Response
Dear Reviewer,
We thank the referee for reading and evaluating the submitted paper.
1. There are not so much data in the main text, it is unnecessary to create a supplementary file for the rest of the data.
Response: With regard to comment number 4, which will be addressed in more detail below, we have carried out additional experiments to determine EC50 values, which were inserted into the main text. This has also led to additional control experiments which are suitable for the supplemental section. Furthermore, we added imaging data via AFM in response to comment number 3, and also chose to add additional images of cell "rounding" for the 1xSWL-DNA and 2xSWL-DNA structures as supporting data. In addition, Table A1 containing the sequences of DNA single-strands was shifted to the main text. The rest of the data was kept in the supplement since it only shows mostly control experiments .Since the main manuscript now contains 5 images, with a further 8 in the supplemental, we feel as though the story is more concisely presented with both a main and supplemental section.
2. Natural ephrin ligands are promiscuous, then why trivalent DNA nanostructure was chosen for this study?
Response: In the case of this study, our choice of a DNA structure enabling trivalent (or generally oligovalent) binding is connected to the necessity for receptor multimerization and clustering, rather than due to the promiscuity of ligands. Our use of the EphA2-specific SWL peptide enabled us to avoid the issue promiscuity. Our goals were to (a) exclusively target EphA2 receptors, and (b) examine the differences between monovalent stimulation (SWL and 1xSWL-DNA), promotion of dimerization through bivalent stimulation (2xSWL-DNA) and promotion of the minimal cluster of 3 EphA2 receptors (3xSWL-DNA) on EphA2 pathways. Therefore, it was advantageous to use the trivalent DNA structure as our basic template, in combination with the EphA2-specific SWL peptide.
3. It seems discussion for the preparation of the DNA nanostructure was ignored in the manuscript, which is odd since this paper is about this DNA nanostructure. Furthermore, it is better to do an AFM characterization of the DNA nanostructure.
Response: Thank you for this suggestion. We included an extra paragraph on the preparation of DNA trimers in section 4.1. High resolution imaging of the DNA structure via AFM characterization is difficult because of its small size. We could make AFM images of structures in that size range (approx. 5-10 nm in diameter) but we could not resolve single DNA arms with our AFM. An image of structures can be found in the supplement in Figure A2.
4. Figure 3 showed only qualitative data of anticancer efficacy, it is better to do more concentration gradients and calculate the IC50 of each formulation.
Response: We truly appreciate this suggestion from the referee, since it has greatly strengthened our manuscript. In carrying out the recommended experiments to quantitatively determine EC50 values, we have found out our structures are even more potent than we had qualitatively estimated based on the original analysis. The results of these experiments are presented in Figure 4 and in the associated texts in sections 2.2 and 3. In particular, we had qualitatively estimated an effective potency in the 5-10 µM range based on binding curves and previous ELISA data, but now know that this is really closer to 20 nM - remarkably more than a factor of 100 improvement. Additionally, we now see that the addition of already one peptide to DNA trimers improved the efficacy compared to the SWL peptide by itself. Doing these experiments enabled us to get a greater insight in the effect of our constructs and it strengthens the paper! The new experiments as well as a more in-depth of how these discussions pertain to the overall point of the paper are addressed in greater detail in the text.
Round 2
Reviewer 1 Report
The authors have appropriately addressed my comments. I recommend the manuscript for publication.
Reviewer 2 Report
The revised manuscript looks good, it is now suitable for publication.